# Tracing the transitions from pluripotency to germ cell fate with CRISPR screening

Jamie A. Hackett [1,2,3], Yun Huang [1,3], Ufuk Günesdogan[1,3,4], Kristjan A. Gretarsson[2], Toshihiro Kobayashi[1,3,5] & M. Azim Surani[1,3]

Early mammalian development entails transit through naive pluripotency towards post-implantation epiblast, which subsequently gives rise to primordial germ cells (PGC), the founding germline population. To investigate these cell fate transitions, we developed a compound-reporter to track cellular identity in a model of PGC specification (PGC-like cells; PGCLC), and coupled it with genome-wide CRISPR screening. We identify key genes both for exit from pluripotency and for acquisition of PGC fate, and characterise a central role for the transcription regulators $Nr5a2$ and $Zfp296$ in germline ontogeny. Abrogation of these genes results in widespread activation ($Nr5a2^{-/-}$) or inhibition ($Zfp296^{-/-}$) of WNT pathway factors in PGCLC. This leads to aberrant upregulation of the somatic programme or failure to activate germline genes, respectively, and consequently loss of germ cell identity. Our study places $Zfp296$ and $Nr5a2$ as key components of an expanded PGC gene regulatory network, and outlines a transferable strategy for identifying critical regulators of complex cell fate decisions.

[1] Wellcome Trust/Cancer Research UK Gurdon Institute, University of Cambridge, Tennis Court Road, Cambridge CB2 1QN, UK. [2] Epigenetics and Neurobiology Unit, European Molecular Biology Laboratory (EMBL), via Ramarini 32, 00015 Rome, Italy. [3] Department of Physiology, Development and Neuroscience, University of Cambridge, Downing Street, Cambridge CB2 3DY, UK. [4] Department of Developmental Biology, University of Göttingen, Göttingen Center for Molecular Biosciences, Justus-von-Liebig Weg 11, 37077 Göttingen, Germany. [5] Present address: Center for Genetic Analysis of Behaviour, National Institute for Physiological Sciences, 5-1 Higashiyama Myodaiji, Okazaki, Aichi 444-8787, Japan. These authors contributed equally: Jamie A. Hackett, Yun Huang and Ufuk Günesdogan. Correspondence and requests for materials should be addressed to J.A.H. (email: jamie.hackett@embl.it) or to M.A.S. (email: a.surani@gurdon.cam.ac.uk)

The germ cell lineage generates the totipotent state and transmits heritable genetic and epigenetic information to the next generation. Robust specification of primordial germ cells (PGC), the precursors of sperm and eggs, is therefore a critical developmental event to ensure the propagation of a species. Approximately 30 specified PGCs are detected in mouse embryos at embryonic day (E) 7.25, which arise from 'competent' epiblast precursor cells in response to BMP and WNT signalling[1,2]. The specification of PGCs is accompanied by induction of key germ-cell genes, repression of the nascent somatic-mesodermal programme, and widespread epigenetic remodelling, including global DNA demethylation[3,4].

PGC specification follows WNT-dependent induction of the primitive streak/mesodermal gene *Brachyury (T)*, which in mice promotes activation of key PGC specifiers; *Blimp1*, *Prdm14*, *Cbfa2t2* and *Ap2γ*[5–8]. These transcription factors form a self-reinforcing network that feeds back to suppress other WNT/BMP-induced mesodermal genes (including *T*), thereby repressing the ongoing somatic programme in early PGCs[9]. *Blimp1* and *Prdm14* additionally activate germline-specific genes and initiate epigenome resetting, with mutation of either gene resulting in loss of PGCs by E12.5 [10,11]. The broader gene regulatory network that controls PGC ontogeny has however been relatively uncharted, due to the absence of unbiased functional approaches and the challenges of analysing the limited number of nascent PGCs in embryos.

The recent development of an in vitro model of PGCs, termed PGC-like cells (PGCLC), now facilitates molecular studies of the specification and developmental events of the germ cell lineage[12,13]. PGCLC are derived by inducing naive embryonic stem cells (ESC) that are equivalent to the inner cell mass (ICM) of blastocysts (E3.5–E4.5), towards competent epiblast-like cells (EpiLC), which closely resemble pre-gastrulation mouse epiblast (E5.5–E6.5)[14,15]. EpiLC can in turn be induced to undergo specification as PGCLC in response to BMP and WNT. Specified PGCLC are equivalent to migratory PGCs in vivo (E8.5–E10.5)[12], and have the potential to develop to mature functional gametes[13,16]. This model is therefore appropriate to investigate the inherent regulatory mechanisms of nascent germ cells, and the preceding developmental transitions from ICM, through post-implantation development and PGC specification.

The advent of genome-wide CRISPR screening has enabled unbiased interrogation of recessive gene function in a wide spectrum of biological contexts[17–19]. We reasoned that by designing appropriate reporters, CRISPR screening could be adapted to identify genes involved in controlling sequential cell fate decisions during lineage-specific differentiation. Specifically, by employing the PGCLC model we embarked on identification of genes that are important for (i) maintenance of naive pluripotency, (ii) transition to the germline competent state (EpiLC) and, (iii) specification of the PGC lineage. We further investigate novel candidates to reveal their key role and mechanistic function during nascent germ cell development. From a broader perspective, we demonstrate that unbiased CRISPR screening can be adapted to probe the genetic basis of complex multi-step developmental processes, using the germline lineage as a paradigm.

## Results

### Compound reporter for developmental transitions to PGC fate.
We set out to design a reporter system that can distinguish between successive cell identities during the developmental transitions from naive pluripotency to specified PGC fate. Single-cell RNA-seq data revealed that *Stella* (also known as *Dppa3*) is expressed in the naive pluripotent ICM but is rapidly downregulated in post-implantation epiblast, with re-activation

occurring specifically in nascent PGCs (from E7.5)[20]. In contrast *Esg1* (also known as *Dppa5*) is also expressed in the ICM, but maintains high expression during post-implantation development, and subsequently becomes strongly downregulated in early PGCs; with low germline expression reacquired later (>E9.5) (Fig. 1a). To exploit the mutually exclusive expression of these genes, we generated an ESC line with compound *Stella*-GFP and *Esg1*-tdTomato (SGET) reporters. In this 'traffic-light' system naive pluripotent cells are double-positive (yellow), post-implantation epiblast cells are *Esg1*-positive (red) and nascent PGCs are *Stella*-positive (green). We monitored SGET expression during mouse development by tetraploid complementation and chimera formation, and observed strong double-activation in naive pluripotent epiblast at E4.5 (Fig. 1b). This resolved to single *Esg1*-tdTomato activity in E6.0 epiblast, with both reporters subsequently silenced in somatic tissues by E7.0. Importantly *Stella*-GFP is specifically reactivated in PGCs by E8.5, until E12.5, with *Esg1*-tdTomato additionally showing weak expression in later PGC stages (Fig. 1b). Thus, SGET faithfully recapitulates expression of the endogenous genes during the transitions towards PGC fate.

Next, we examined SGET activity during in vitro specification of PGCLC, which initiates from naive ESC and transitions through 'competent' EpiLC[12]. SGET ESC were largely double-positive ($SG^+ET^+$), but resolved to *Esg1*-only expression ($SG^-ET^+$) in >97% of cells upon induction into EpiLC. In contrast, nascent PGCLC (day 2) reactivated *Stella* while concomitantly repressing *Esg1* ($SG^+ET^-$), with later stage PGCLC (day 6) exhibiting low *Esg1* expression ($SG^+ET^{low}$) (Fig. 1c). PGCLC carrying SGET therefore recapitulate the in vivo dynamics. To functionally validate the SGET reporter system, we generated ESC with a mutation in the key PGC-specifier *Blimp1* (Supplementary Fig 1A), which resulted in a significant reduction (up to 2.8-fold) in the efficiency of PGCLC generation (Fig. 1d; Supplementary Fig 1B). The remaining $Blimp1^{-/-}$ PGCLC exhibited aberrant gene expression (Supplementary Fig 1C), consistent with in vivo studies where a fraction of nascent 'PGCs' remain but with an altered transcriptome[10]. The SGET reporter system is thus ideal for monitoring successive changes in cell identity, from naive ESC to specified PGCLC.

### CRISPR screen identifies key genes for naive ESC.
We introduced a single-copy of *Cas9* into SGET ESC (Supplementary Fig 1D), and subsequently infected this line with an integrating lentiviral library of exon-targeting guide RNAs (gRNA)[21], in independent biological replicates. Statistical analysis of gRNA frequency in the ESC population, relative to the initial frequency, reveals essential genes because their cognate gRNAs become depleted concomitant with cell loss. We reasoned that by flow sorting SGET cells that successfully acquire each sequential fate during PGCLC specification, we could identify essential genes for each transition by comparing gRNA frequency to the preceding cell population (Supplementary Fig 1E). Accordingly, this approach filters out genes with a role in prior fate transitions and reveals the critical regulators for each stage of multi-step differentiation events, based on functional requirement.

Following introduction of the CRISPR library, we identified 627 genes that lead to a loss of naive ESC maintained in 2i/LIF when knocked-out ($p < 0.01$ in all independent replicates using the relative ranking algorithm (RRA) in the MAGeCK tool[22]) (Fig. 2a). Gene ontology (GO) terms for these genes were highly enriched for fundamental biological processes such as ribosome biogenesis (Bonferroni adjusted $p = 2.88 \times 10^{-27}$; e.g. *Rps5*) and protein translation (Bonferroni adjusted $p = 4.25 \times 10^{-30}$; e.g. *Eif6*), supporting the efficacy of the library (Supplementary Fig 2A). Notably ESC carrying mutations for the core

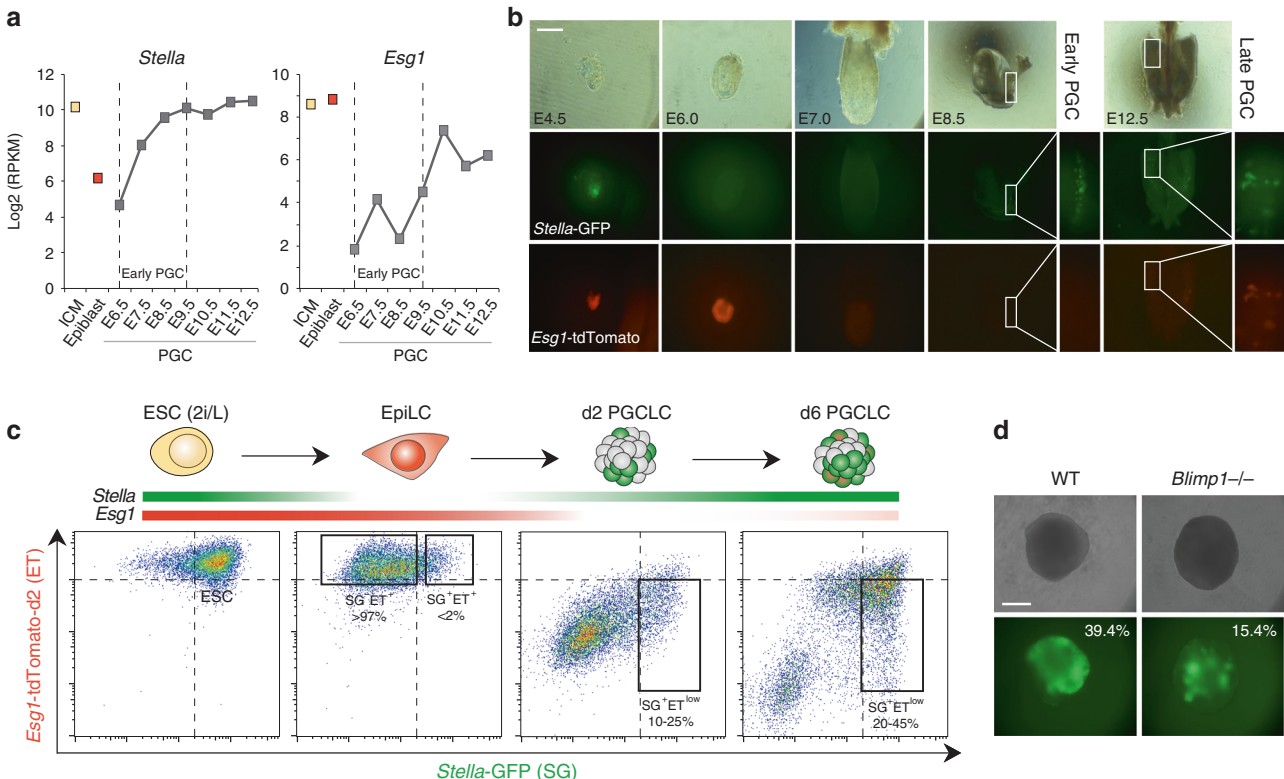

**Fig. 1** The SGET reporter for tracking cell fate transitions toward PGCs. **a** Single-cell RNA-seq analysis showing reciprocal gene expression patterns of endogenous *Stella* and *Esg1* during early development (ICM; E3.5 or Epiblast; E6.5) and in primordial germ cells (PGC). **b** Chimera and tetraploid complementation assays confirm faithful expression of the *Stella*-GFP (green) and *Esg1*-tdTomato (red) SGET reporter during early development and in PGCs. **c** Representative FACS analysis of SGET activation during in vitro cell fate transitions of ESC into EpiLC and subsequently PGCLC. **d** Representative images of impaired PGCLC induction from *Blimp1*−/− SGET ESC. Numbers indicate percentage of SG+ETlow PGCLC at day 5 as determined by FACS. Scale bar(s): 200 μm

pluripotency factors *Oct4* and *Sox2* were also highly depleted, consistent with their essential role in propagating pluripotent ESC[23] (Fig. 2a, b; Supplementary 2B). The majority of naive pluripotency genes were not depleted however, which is in line with the capacity of 2i/LIF media to buffer against a single genetic perturbation to the naive pluripotency network (Fig. 2b)[24]. Importantly, this enables the role of such naive pluripotency genes to be examined in PGC specification, where they are also typically expressed, since knockout cells remain in the population. Exceptions however are *Nanog* and *Tfcp2l1*, which are likely depleted owing to a proliferation disadvantage upon knockout in ESC. In contrast to the many depleted genes, we observed that only 21 genes become enriched in the population upon knockout, primarily corresponding to tumour suppressors (Supplementary Fig 2B & C). Among these *Trp53* (*p53*) is the top hit, suggesting that *p53*-mutant mouse ESC acquire a major selective advantage, similar to recent reports in human ESC[25].

We next compared the genes essential for naive ESC (in 2i/LIF) with those essential for ESC in conventional serum/LIF culture, which represents an alternative pluripotent state[23]. Using datasets from the same CRISPR library[21], we observed good overlap (62%) between ESC conditions, but noted 38% of genes appear to be critical under only specific pluripotent culture parameters (Fig. 2c). Among the 2i/LIF specific genes were *Ogdh* and *Dlst*; two enzymes that mediate α-ketoglutarate metabolism which feeds into the TCA cycle. We used siRNAs to test acute loss of *Ogdh* and observed a significant reduction of proliferation and/or increased cell death in 2i/LIF ESC but not serum/LIF ESC (Fig. 2d). We also tested *Txn1*, which was scored as essential in both pluripotent ESC states, and found an equivalent cell

reduction in 2i/LIF and serum/LIF (Fig. 2d). These data suggest ESC in distinct pluripotent states rely, in part, on distinct genetic networks, with ESC in 2i/L uniquely reliant on *Ogdh* metabolism of α-ketoglutarate for example. More generally, establishing this mutant SGET ESC population enables screening for functional regulators of EpiLC, and subsequently PGCLC, without confounding general survival genes.

**Acquisition of post-implantation epiblast fate**. We next used fluorescence-activated cell sorting (FACS) to isolate EpiLC that successfully transited to the SG−ET+ epiblast state, as well as cells that maintained *Stella* expression (SG+ET+) (<2%), indicative of failure to exit naive pluripotency. Analysis of this SG+ET+ EpiLC population revealed enrichment for 21 candidate genes with potential intrinsic roles in dissolution of naive pluripotency (*p* < 0.01 using the RRA[22]), including the two prototypical regulators *Tcf3* and *Zfp281* (Fig. 2e; Supplementary Fig 3A)[26,27]. The candidates additionally included the epigenetic regulator *Dnmt1* and also *Rest*, which are known to be important during early differentiation[28,29].

To test the role of novel candidates in driving exit from pluripotency we generated ESC knockouts for *Zmym2* (also known as *Zfp198*), *FoxP1*, *Uchl5* and *Zfp281* as a positive control, using CRISPR targeting. Consistent with their failure to repress *Stella*-GFP during EpiLC formation, mutant EpiLC were also impaired in activation of key epiblast markers *Fgf5*, *Dnmt3b* and *Otx2*, while their proliferation appeared unaffected (Fig. 2f; Supplementary Fig 3B-C). We observed both a delay in activation of epiblast genes and a reduction in their absolute levels in

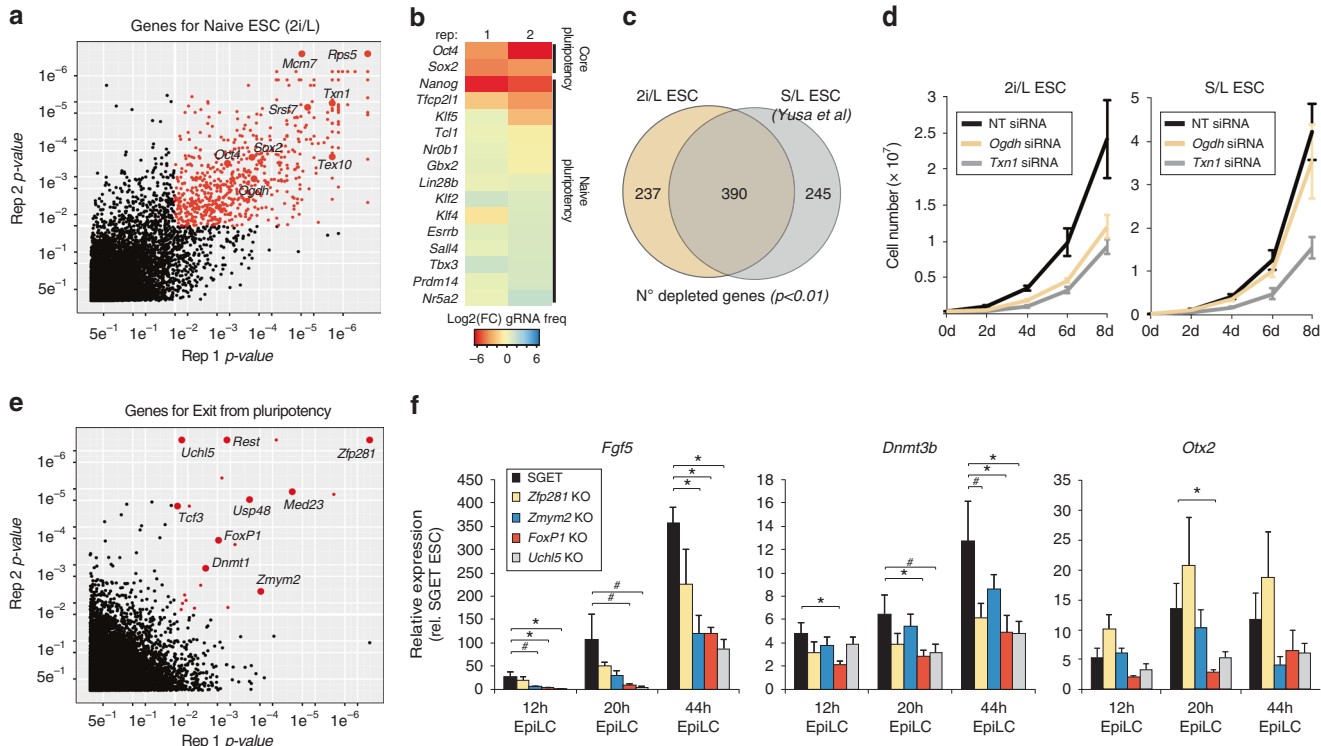

**Fig. 2** Identification of important genes for ESC and EpiLC induction. **a** Scatter plot showing essential genes for ESC propagation in 2i/LIF (red datapoints) as determined by a CRISPR screen. **b** Heatmap showing mean fold change (FC) in normalised frequency of gRNAs targeting core- and naive- pluripotency genes in ESC relative to frequency in starting gRNA library. Reduced frequency indicates a functionally important gene for ESC in 2i/LIF. **c** Venn diagram intersecting significant depleted genes in 2i/LIF ESC (2i/L) with serum/LIF (S/L) maintained ESC. **d** Proliferation of ESC following siRNA knockdown of *Ogdh* or *Txn1* in ESC maintained in 2i/L or S/L culture conditions. **e** Scatter plot showing significantly enriched gene knockouts in SG+ET+ EpiLC that have failed to exit naive pluripotency. **f** Gene expression showing impaired activation of key epiblast markers in EpiLC carrying knockouts for candidate exit from pluripotency regulators. Significance was determined using a one-tailed *t*-test; *p < 0.05; #p < 0.1. Error bars show s.e.m. of triplicate independent experiments

knockout lines, despite high levels of powerful differentiation-inducing FGF and ACTIVIN in the culture medium. Notably, the *Zmym2*-, *FoxP1*-, *Uchl5*- mutant lines exhibited expression defects comparable with the bona fide exit from pluripotency regulator *Zfp281*, supporting an important role for them in the timely acquisition of post-implantation cell identity. Consistently all three candidates are transcriptionally upregulated during EpiLC induction (Supplementary Fig 3D).

**Candidate genes for specification of PGC fate**. We next focused on genes involved in mouse germline specification by inducing PGCLC from competent SG−ET+ EpiLC. In order to obtain sufficient numbers of PGCLC for coverage of the gRNA library, we optimised and scaled-up the induction (Supplementary Fig 4A). Specified SG+ETlow PGCLC were FACS purified at day (d) 6 of differentiation to allow time for depletion of cells carrying mutations in critical germline genes. Because PGCLC numbers remained a limiting factor however, we relaxed the candidate threshold to p < 0.05, while retaining a p < 0.01 in at least one replicate, to account for increased noise (Supplementary Fig 4B). This resulted in identification of 23 candidate genes involved in specification and/or development of nascent PGC. The frequency of gRNAs targeting these genes was highly depleted in PGCLC relative to preceding EpiLC (Fig. 3a), while negative control gene families were not depleted (Supplementary Fig 4C). Moreover, genes with an established critical role in PGC specification, such as *Blimp1*, *Cbfat2t2* and *Prdm14*, also showed marked reduction in gRNA frequency in PGCLC, supporting the efficacy of the

screen (Fig. 3a). These genes narrowly failed to meet significance across all replicates however; *Cbfa2t2* scored p-values of p = 0.00006 and p = 0.22 for example. We also noted that several pluripotency genes (*Nr5a2*, *Esrrb* and *Sall4*) were depleted specifically from PGCLC (Supplementary Fig 4C), suggesting that some genes typically linked with pluripotency have a potentially important function in the germline.

To refine our candidate list we examined their expression profile during induction of PGCLC from the transduced SGET ESC by RNA-seq. This confirmed that SG+ETlow PGCLC activate high levels of key PGC markers such as *Blimp1*, *Stella*, *Itgb3*, *Nanos3* and *Prdm14*, as well as repressed *Uhrf1* and *Klf4*, supporting their authentic germline identity (Supplementary Fig 4D). The dynamics of candidate gene expression revealed four broad clusters, with cluster 1 and 4 corresponding to genes that are expressed relatively stably throughout ESC to PGCLC transition; the latter is nonetheless only just above the detection threshold implying it may filter out false-positives. In contrast, cluster 2 and 3 reflected dynamically regulated genes that are activated in PGCLC, with cluster 3 corresponding to genes fully silenced in EpiLC before strong germ cell re-activation, and including known PGC specifiers (Fig. 3b). From these, we chose to focus on *Zfp296* and *Nr5a2*, since they rank 3 and 7 overall in the screen, and exhibit comparable PGCLC expression dynamics to *Prdm14* and *Blimp1* (Fig. 3b; Supplementary Fig 4D). *Nr5a2* encodes an orphan nuclear receptor previously associated with reprogramming towards pluripotency, and also has a role in extra embryonic development, which leads to embryonic lethality when

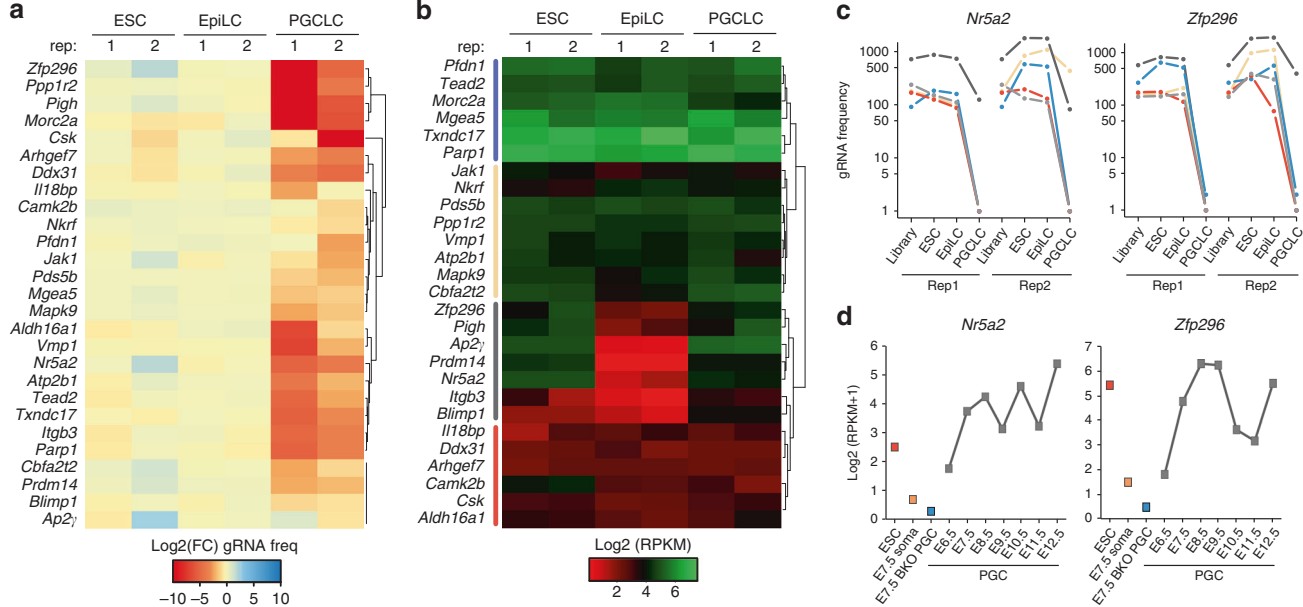

**Fig. 3** Candidate genes for primordial germ cell fate. **a** Heatmap showing mean fold-change (FC) in normalised frequency of gRNAs that target candidate PGC regulators, relative to their frequency in the preceding cell state. **b** Log2 RPKM gene expression dynamics of candidate PGC regulators during induction of SGET ESC into d6 PGCLC in the screen. **c** Normalised frequency of individual gRNAs targeting *Nr5a2* and *Zfp296* during induction of PGCLC. **d** Expression dynamics of *Nr5a2* and *Zfp296* during in vivo formation of PGCs by single-cell RNA-seq. *Blimp1*$^{-/-}$ PGC (BKO) and soma are shown for reference

abrogated[30]. *Zfp296* encodes a zinc-finger protein that was among the original 20 Yamanaka factors, and has recently been shown to influence heterochromatin and induced pluripotent stem cell (iPSC) generation[31–33].

Inspection of the individual gRNAs that target *Nr5a2* and *Zfp296* revealed a strong and reproducible reduction of their frequency specifically in PGCLC, but not in the preceding developmental transitions (Fig. 3c). Moreover, single-cell RNA-seq indicates both genes are highly upregulated in nascent PGC in vivo at the very onset of specification, but not in adjacent somatic cells (Fig. 3d). Both *Nr5a2* and *Zfp296* are also strongly upregulated in early human PGCs (Supplementary Fig 4E), and fail to be activated in *Blimp1*$^{-/-}$ PGC that are destined to diverge from the germline lineage (Fig. 3d). The striking upregulation of these genes during incipient PGC specification, coupled with their high rank in the functional screen, prompted us to investigate their role in detail.

**Nr5a2 and Zfp296 regulate germline ontogeny**. We used CRISPR editing to generate frame-shifted null-alleles of *Nr5a2* or *Zfp296* in SGET ESC. *Nr5a2*$^{-/-}$ ESC proliferated normally, expressed comparable levels of naive pluripotency genes, and exhibited indistinguishable morphology from parental and matched WT ESC derived from the same targeting process. However upon induction of PGCLC, we observed a significant reduction in specification efficiency of *Nr5a2*$^{-/-}$ cells at d2 relative to WT controls (WT 19.8% ± 8.9; KO 6.4% ± 2.3) (Fig. 4a). The impaired induction of *Nr5a2*-mutant cells was further reflected in d6 PGCLC, and across multiple independent knockout lines (WT 44.7% ± 4.0; KO 17.4% ± 4.6), suggesting an important role for *Nr5a2* in PGC specification (Fig. 4a; Supplementary Fig 5A). We next tested *Zfp296*$^{-/-}$ ESC lines, which also exhibited normal morphology and pluripotent gene expression profiles, albeit with modestly reduced proliferation rate. Upon induction, PGCLCs were specified from *Zfp296*-mutant cells at apparently normal

efficiency at d2 (WT 14.9% ± 3.2; KO 16.2% ± 2.2), as judged by the SGET reporter. However, by d6 we observed a highly significant depletion of PGCLC in multiple mutant lines (WT 43.1% ± 3.5; KO 8.9% ± 2.8), suggesting the effects of *Zfp296* abrogation on PGCLC are progressive and manifest after specification (Fig. 4b). More generally, these data indicate that loss-of-function of either *Nr5a2* or *Zfp296* leads to a significant impairment of germ cell development.

Despite the dramatic reduction in numbers, both mutant lines give rise to some putative PGCLC. To examine the identity of these cells, and their developmental trajectory, we performed RNA-seq. Unbiased hierarchical clustering revealed that *Nr5a2*$^{-/-}$ EpiLC were interspersed with WT, but that PGCLC were clustered according to genotype, implying transcriptional differences in the absence of *Nr5a2* primarily arise upon induction of germline fate (Supplementary Fig 5). To investigate this further we used principal component analysis (PCA), which confirmed that mutant EpiLC cluster closely with WT but that *Nr5a2*-null PGCLC progressively diverge from WT controls during their development (Fig. 4c). By d6 a strong distinction is apparent. This is consistent with *Nr5a2*$^{-/-}$ EpiLC being competent for the germline lineage but undergoing an abnormal developmental trajectory upon PGCLC specification, which coincides with the timing of *Nr5a2* upregulation in WT cells.

Global transcriptome clustering of *Zfp296*-null cells indicated that precursor EpiLC were distinct yet highly comparable between WT and knockout lines (Supplementary Fig 5). Indeed, PCA showed *Zfp296*$^{-/-}$ EpiLC cluster closely with WT. In contrast, d2 mutant PGCLC exhibited a significantly different state from WT counterparts, that is closer to precursor EpiLC and thus apparently impaired in its developmental path towards germline fate, which is consistent with the subsequent loss of PGCLC numbers (Fig. 4d). The remaining *Zfp296*-mutant PGCLC at d6 appeared to partially re-establish the expected global profile, albeit with greater transcriptome variance between lines than WT. This implies that despite widespread

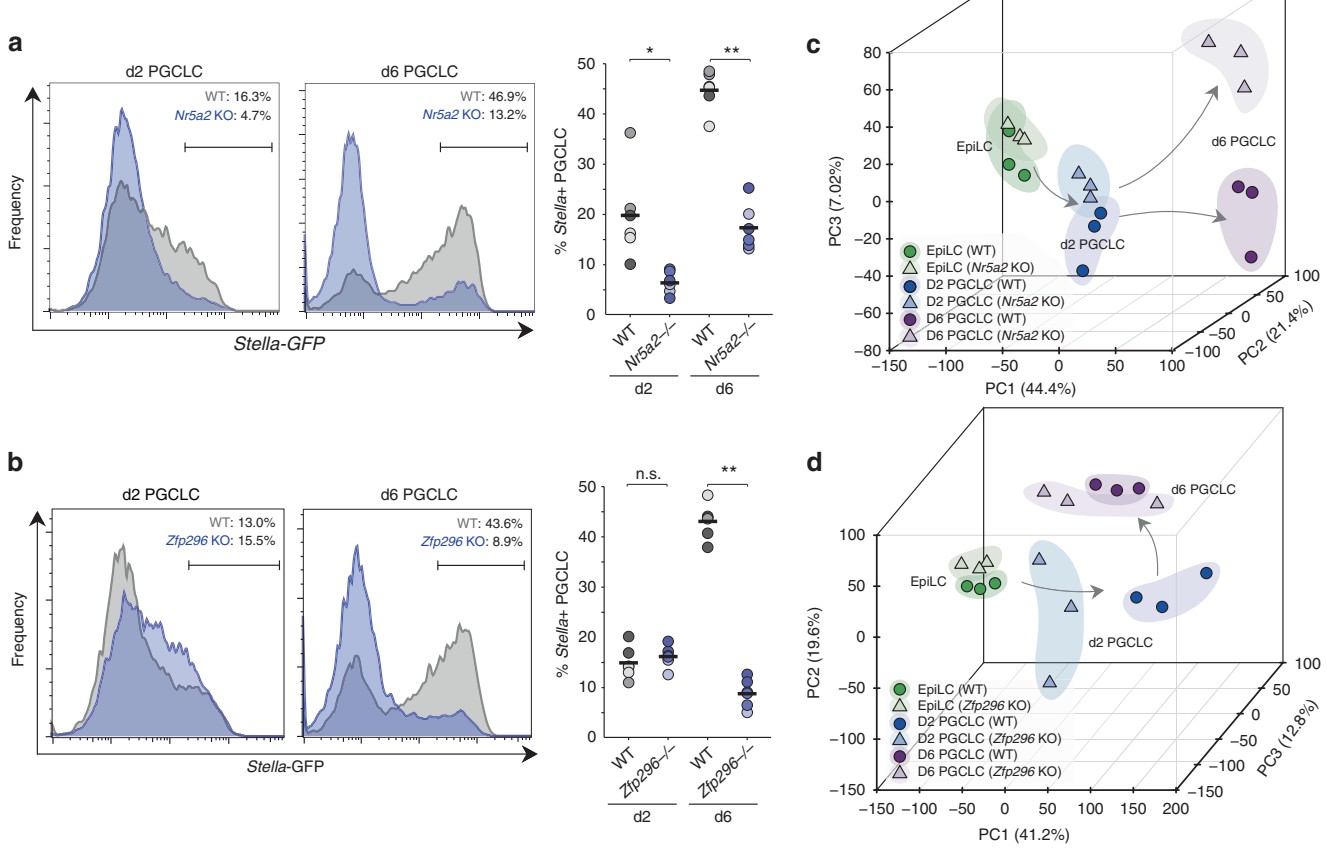

**Fig. 4** *Nr5a2* and *Zfp296* are key regulators of germ cell development. **a** Representative FACS plot showing impaired induction of PGCLC in *Nr5a2*-knockout (KO) cells. Shown right is replicate quantifications of independent WT and KO lines. Independent lines are colour-matched. Bar indicates mean value. **b** Representative FACS of PGCLC in *Zfp296*-knockout SGETs, with replicate quantification shown right. **c** Principal component analysis (PCA) showing the developmental trajectory of independent *Nr5a2*$^{-/-}$ and matched-WT SGET lines during induction of PGCLC, based on global transcriptome. **d** PCA of independent *Zfp296*$^{-/-}$ and matched-WT SGET transcriptomes. *$p < 0.05$; **$p < 0.01$

transcriptional mis-regulation, a minor proportion of mutant PGCLC can overcome the effects of *Zfp296* abrogation, and give rise to putative germ cells (Fig. 4d). Overall, these data suggest that loss of *Nr5a2* or *Zfp296* has a dramatic influence on PGC development.

**Nr5a2 and Zfp296 converge on reciprocal regulation of WNT.** To investigate the mechanisms that impair development of mutant PGCLC and underpin their aberrant transcriptomes in more depth, we identified differentially expressed genes (adjusted $p < 0.05$; >2 FC) in EpiLC and PGCLC. This revealed that downregulation of naive pluripotency genes and upregulation of post-implantation markers proceeded similarly between *Nr5a2*$^{-/-}$ and WT EpiLC counterparts (Fig. 5a). Moreover, there was appropriate upregulation of early PGC markers such as *Nanos3*, *Stella*, *Blimp1* and *Prdm14* in d2 *Nr5a2*$^{-/-}$ PGCLC, that was at least equivalent to WT, albeit with a modest failure to upregulate *Nanog*. However, we noted d2 *Nr5a2*-null PGCLC exhibited a highly significant overexpression of WNT pathway genes and targets, including *Cdx2*, *T* and *Mixl1* (Fig. 5a). This was further reflected in gene ontology (GO) analysis of differentially expressed genes at d2, which highlighted canonical WNT signalling pathway (Bonferroni adjusted $p = 0.017$) and pattern specification process (Bonferroni adjusted $p = 0.014$) as top hits (Table S1). We confirmed hyperactivation of key WNT pathway genes *T* and *Wnt3* in early *Nr5a2*$^{-/-}$ PGCLC using qRT-PCR on

independent replicates (Fig. 5b). Strong overexpression of the master mesoderm regulator *T* is predicted to override its role in germ cell induction and divert cells toward a somatic mesendoderm programme. Consistently, the surviving *Nr5a2*$^{-/-}$ PGCLC at d6 exhibit acute activation of mesendoderm regulators such as *Tbx4*, *Hey1* and *Pou2f3* (Fig. 5a). This collectively suggests that the absence of *Nr5a2* leads to aberrant transduction of WNT signalling in nascent PGCLC, which in turn leads to upregulation of mesendoderm master-regulators and consequently de-repression of a somatic programme. It is possible that the aberrant activation of somatic genes in the absence of *Nr5a2* is a contributing factor that drives the majority of prospective PGCLC out of the germline programme, leading to reduced efficiency of specification.

We next analysed gene expression in *Zfp296*-null EpiLC, which revealed that pluripotency genes including *Nanog*, *Esrrb* and *Tfcp2l1* were repressed as expected, while EpiLC markers such as *Fgf5* and *Dnmt3b* were appropriately upregulated. In contrast mutant PGCLC at d2 exhibited a clear signature of mis-expressed genes (Fig. 5a). Remarkably, GO analysis again suggested this primarily corresponded to WNT targets, but in contrast to hyperactivation of WNT in *Nr5a2*$^{-/-}$ PGCLC, *Zfp296*-null PGCLC exhibited a severe block of WNT-associated gene activation; the top enrichment is negative regulation of canonical WNT signalling (Bonferroni adjusted $p = 0.0064$). Among all genes *T* is the most significantly downregulated in early *Zfp296*-mutant PGCLC, with *Cdx2*, *Notum* and *Dkk1* also being among

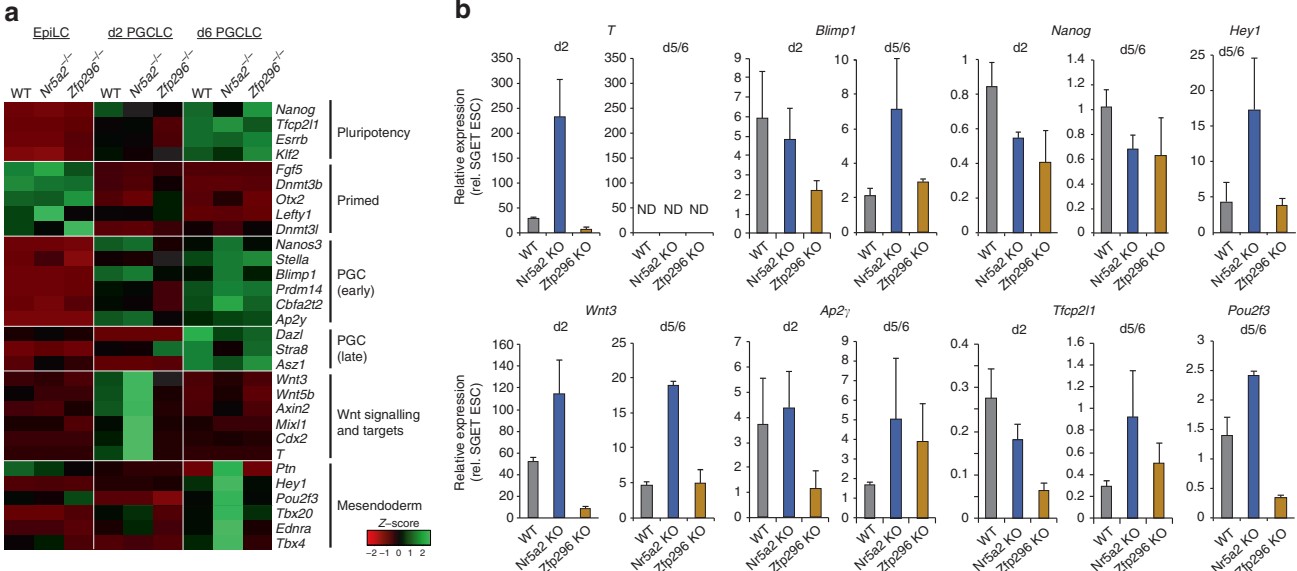

**Fig. 5** Compound mis-regulation of WNT and germ cell genes in the absence of *Nr5a2* and *Zfp296*. **a** Heatmap showing change in mean expression profile of selected genes from triplicate independent WT, *Nr5a2⁻/⁻* or *Zfp296⁻/⁻* lines during PGCLC induction by RNA-seq. **b** Quantitative validation of gene expression changes in PGCLC from WT, *Nr5a2⁻/⁻* and *Zfp296⁻/⁻* lines by independent qRT-PCR experiments. Error bars show s.e.m. of duplicate biological experiments, each comprising at least twelve independent replicate inductions. ND not detected

the top 20. The absence of appropriate level WNT signalling could manifest as impaired activation of critical germline genes that are WNT and/or *T* targets[5]. Indeed, *Zfp296*-null PGCLC at d2 exhibited a block in *Blimp1*, *Prdm14 Cbfa2t2a* and *Ap2γ* activation (Fig. 5a). qRT-PCR analysis independently confirmed highly impaired *T* and *WNT* genes in *Zfp296*-mutant PGCLC and concomitant significant reduced activation of germline regulators *Blimp1* and *Ap2γ* (Fig. 5b). This suggests that absence of *Zfp296* establishes a reciprocal developmental failure relative to deletion of *Nr5a2*, with both genes converging on regulation of appropriate WNT signalling in nascent PGCLC. In the case of *Zfp296*, this appears to manifest in a delayed but dramatic loss of PGCLC by d6. Notably, the SG⁻ET⁻ cells from both *Nr5a2* and *Zfp296* embryoids, which have putatively acquired somatic fate, exhibited comparable expression of lineage-specific markers *Sox17*, *Hoxa1* and *Sox7* relative to wild type (Supplementary Fig 6A). This implies that the loss of function for both genes does not impede the exit from the pluripotent state towards somatic fates, but these cells are impaired specifically in their capacity to form PGCs.

**Rescue of Nr5a2 and Zfp296 deficiency.** We considered that the impaired specification and aberrant transcriptome of mutant PGCs could be due to impaired epigenetic reprogramming, as reported in *Prdm14⁻/⁻* PGCs[11]. To investigate this, we quantified global DNA methylation erasure and observed that WT, *Nr5a2⁻/⁻* and *Zfp296⁻/⁻* PGCLC all underwent extensive and comparable DNA demethylation (Fig. 6a). This was confirmed by bisulfite pyrosequencing analysis of LINE-1 elements (Supplementary Fig 6B). We therefore turned our attention back to understanding whether the observed mis-regulation of WNT could be driving impaired germline fate by exposing incipient PGCLC to WNT inhibitor (WNTin: XAV939) or WNT agonist (WNTag: Chiron). Addition of WNTin led to strong down-regulation of key PGC genes including *T* and *Wnt3*, as well as *Nanog*, consistent with the effects of impaired WNT transduction in *Zfp296*-null PGCLC (Fig. 6b). Reciprocally, WNTag affected an increase in expression of *T* and *Wnt3* at d2, while also down-regulating *Nanog*, which is in line with the effects of WNT over-

activity in *Nr5a2*-null PGCLC. Indeed, WNTag also elicited overexpression of mesendoderm gene *Hey1* by d6 PGCLC (Fig. 6c). Notably WNTinh led to a significant depletion of specified PGCLC (*p < 0.05*), while partial WNT activation by WNTag also trended towards impaired PGCLC specification (Fig. 6d). These data suggest that precise levels of WNT activity are necessary for optimal PGC specification.

Addition of WNTin to *Nr5a2⁻/⁻* PGCLC rescued hyperactivated *T* and *Wnt3*, demonstrating that their upregulation in *Nr5a2*-mutant PGCLC is likely a direct consequence of de-regulated WNT activity (Fig. 6b). In contrast addition of excess WNTag to *Zfp296*-mutant PGCLC only resulted in modest rescue of aberrantly repressed WNT targets, suggesting that in the absence of *Zfp296*, germ cells cannot respond to WNT activation, even when forced. We therefore attempted to rescue the specification defects by generating mutant PGCLC that carry Doxycycline (DOX)-inducible expression of wild-type *Nr5a2* and *Zfp296*. We induced *Nr5a2* expression specifically during PGCLC induction in a *Nr5a2⁻/⁻* background, which led to a highly significant rescue of PGCLC specification. Moreover, this reimposed suppression of somatic and WNT target genes, implying a direct effect of *Nr5a2* on these pathways (Fig. 6e; Supplementary Fig 6C-D). In contrast, we found expression of *Zfp296* to be highly toxic to both ESC and PGCLC, even at very low background (leaky) levels, suggesting precise control over its expression level is important.

Remarkably DOX-induced *Nr5a2* expression in mutant PGCLC resulted in increased PGCLC induction as compared to wild-type counterparts. This encouraged us to test whether *Nr5a2* could drive germline fate independently of BMP cytokines that are normally requisite. Here, forced expression of *Nr5a2* in wild-type EpiLC led to emergence of SG⁺ET^low PGCLC at >tenfold higher efficiency than background; at levels comparable with forced expression of canonical PGC-specifier *Prdm14*, albeit at low absolute frequency (3.3%) (Fig. 6f). Induction of *Nr5a2* resulted in PGCLC with apparently similar expression profiles as authentic PGCs (Supplementary Fig 6E). This indicates that activation of *Nr5a2* can, at low efficiency, directly induce PGC-like fate from competent EpiLC. More

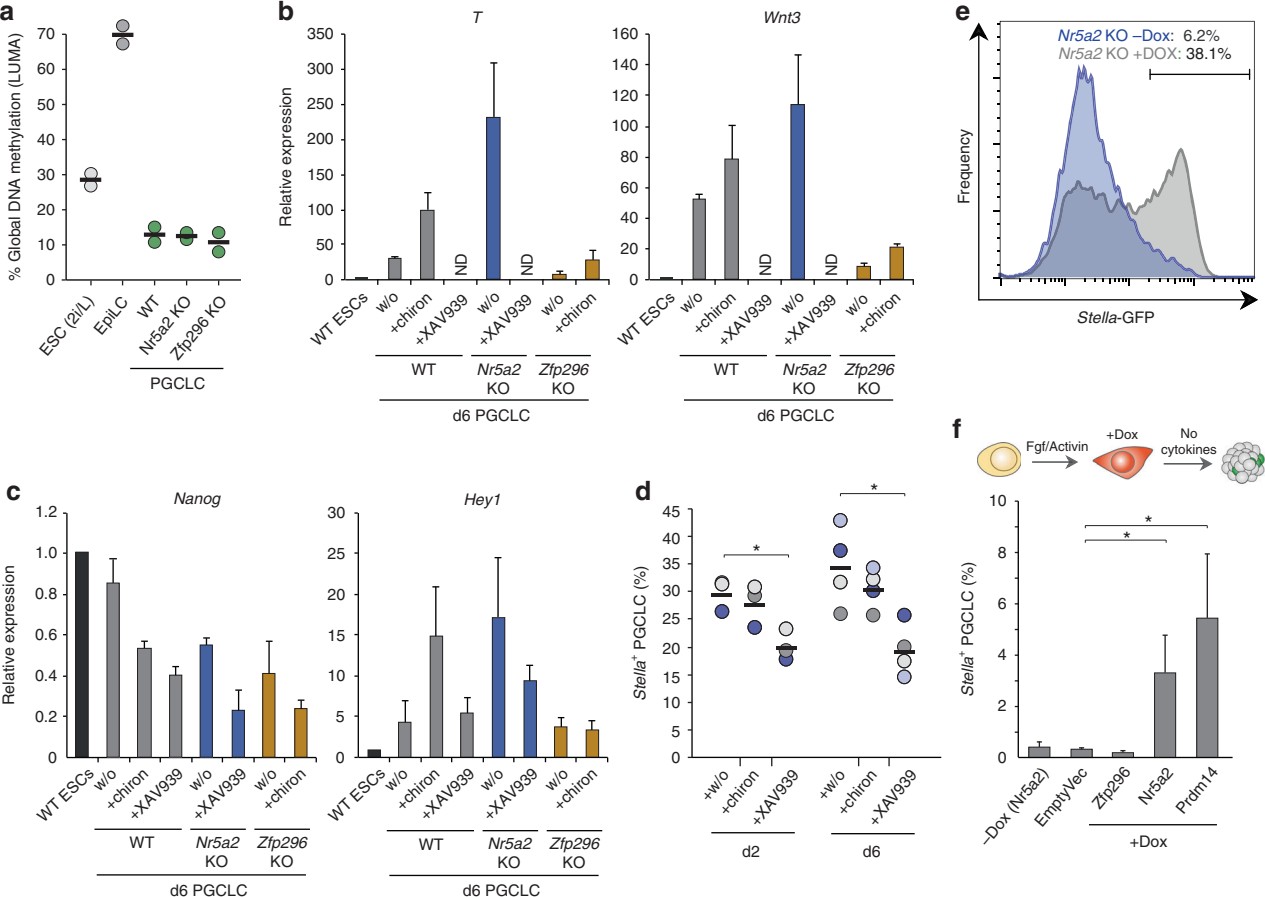

**Fig. 6** Role of WNT and rescue of *Nr5a2* defects. **a** Global levels of DNA methylation reprogramming in WT and mutant PGCLC assayed by LUMA. **b** qRT-PCR analysis showing expression of WNT genes/targets are affected by WNT inhibitor (+XAV939) or agonist (+chiron), similarly to knockout PGCLC. w/o: control without WNT modulation. **c** Expression of *Nanog* and mesendoderm gene *Hey1* by qRT-PCR in WT and mutant PGCLC with WNT modulation. **d** Effects of WNT manipulation on efficiency of PGCLC ontogeny in quadruplicate independent experiments. Bar indicates mean value and *$p < 0.05$. **e** FACS plot showing the *Nr5a2*-mutant PGCLC specification defect at d2 is rescued by Dox-inducible activation of exogenous *Nr5a2* cDNA. Dox is added after EpiLC induction. **f** Percentage *Stella*+ PGCLC induced without cytokines from WT cells upon DOX-inducible expression of the indicated cDNA

generally, our data places *Nr5a2* and *Zfp296* as critical regulators that provide robustness to germline specification by modulating appropriate WNT activity.

## Discussion

Our study outlines a principle for using CRISPR screens to deconvolve the genetic basis of successive cell fate decisions. By establishing a 'traffic-light' SGET reporter system we traced the pathway from pluripotency to germ cell fate, and identified important genes for each transition. This strategy can be transposed to interrogate multi-step transitions towards any lineage for which a faithful compound reporter or isolation method can be developed.

Utilising SGET we identified genes involved in exiting naive pluripotency, including the previously described key regulators *Zfp281* and *Tcf3*[34,35], and further validated roles for novel candidates *Zmym2*, *FoxP1* and *Uchl5*. Among these *Zmym2* expression is anti-correlated with naive pluripotency genes at the single-ESC level[36], and is upregulated when cells are induced to egress from naive pluripotency both in vitro and in vivo (Supplementary Fig 3). Indeed, ZMYM2 is known to directly interact with NANOG[37] where its negative regulatory activity could blunt NANOG function, or it could alternatively function in this role as part of the LSD1 complex[38]. Interestingly the transcription

regulator *FoxP1* is also strongly activated during EpiLC induction and, while it can be alternatively spliced to generate a pro-pluripotency isoform[39], our data implies canonical *Foxp1* isoforms play a role in antagonising naive pluripotency. The potential mechanism of function for the deubiquitinase *Uchl5* is less clear, but it has previously been associated with regulation of the TGF-β/Smad pathway[40]. Future work will be important to characterise the regulatory role of the full repertoire of candidates in control of pluripotent status.

We further exploited the SGET strategy to identify *Zfp296* and *Nr5a2* as key functional genes for the germline lineage and as important parts of an expanded PGC gene regulatory network. Deletion of either *Nr5a2* or *Zfp296* converges on mis-regulation of the WNT pathway, resulting in its aberrant activation or inhibition, respectively, implying that precisely regulated levels of WNT is fundamental for germ cell ontogeny. Indeed, loss of either gene leads to severely disrupted PGCLC development. Of note *Nr5a2* is a reported WNT target[41] and directly interacts with β-catenin[42], raising the possibility it forms a negative-feedback loop with WNT to modulate appropriate signalling levels in PGCs. This must strike a balance between WNT-mediated activation of germ cell regulators such as *Blimp1* and *Prdm14*[5], and the capacity to subsequently repress the WNT co-activated mesodermal programme. By blunting WNT activity, NR5A2 may ensure the threshold is tipped towards reversibility, since in its

absence PGCLC development is impaired and linked with aberrant activation of mesendodermal genes. Functionally, *Nr5a2* was previously shown to be important for reprogramming somatic cells to iPSC but not for pluripotency per se[43–45]. Similarly, PGC undergo reprogramming but do not acquire pluripotency[4], supporting a role for *Nr5a2* in resetting PGCs away from somatic fates. Notably, NR5A2-binding motifs are among the most significantly enriched within germline-associated promoters[46], consistent with a direct functional role for *Nr5a2* on PGC genes.

In contrast to *Nr5a2*, loss of *Zfp296* results in failure to appropriately upregulate WNT in the germline, as well as affecting some TGF-β targets such as *Lefty1*. Underactive WNT in *Zfp296*-mutant PGCLC manifests as impaired activation of early germline genes, including *Blimp1* and *Prdm14*. This in turn leads to dramatic loss of PGCLC. Indeed, a recent report has shown the absence of ZFP296 leads to sterility, confirming the long-term consequences of its loss-of-function on the germline[32]. The molecular mechanism through which *Zfp296* impacts WNT to regulate PGC ontogeny is an important topic for future research. ZFP296 can, however, influence global H3K9 methylation and has been reported to negatively regulate KLF4 activity[32,47]. The latter counteracts murine *Blimp1* expression[48], implying *Zfp296* could contribute to the PGC programme through multiple regulatory routes. Of note, both *Zfp296* and *Nr5a2* are strongly activated in nascent human PGC (Supplementary Fig 4)[49], comparably to their upregulation in mouse PGC, indicating they may have a broad conserved role in mammalian germline development.

Our approach and experimental design is both objective and based on functional outcomes, thereby delineating an alternative strategy to identify critical components of cell-type specific gene regulatory networks, which in the case of germ cells, has previously been challenging. The approach preferentially identifies intrinsic cell fate regulators however; not weak-acting or extrinsic regulators, for at least two reasons. First, the addition of extrinsic signalling at high levels (FGF and ACTIVIN for EpiLC; BMP among others for PGCLC) overcomes weak- or signalling receptor-specific effects by acting on multiple redundant pathways to drive fate transitions. Second, the pooled screen strategy means that knockout cells for a given extrinsically acting gene (e.g. *Wnt3*) can be rescued by paracrine effects from adjacent cells with wild-type alleles. For these reasons, FGF signalling and WNT components were not identified in the exit from pluripotency and PGCLC screens, respectively. Nevertheless, we identified multiple cell-intrinsic candidates. Indeed, recent optimisations to loss-of-function CRISPR libraries[50,51] will minimise false negatives in future iterations, which can arise due to ineffective gRNAs confounding significance scores, as we observe for *Prdm14* for example. In summary, we have used CRISPR screening to identify key regulators for successive cell fate transitions out of naive pluripotency towards the germ cell lineage. In doing so we characterised *Zfp296* and *Nr5a2* as crucial for driving germ cell development by acting to regulate appropriate WNT levels, thereby controlling the balance between activation of PGC genes and silencing of the somatic programme.

## Methods

**ESC culture and PGCLC induction**. ESCs were maintained in N2B27 medium supplemented with 2i (PD0325901 (1 μM) and CHIR99021 (3 μM; both Stemgent), 1000 U/ml LIF (Cambridge Centre for Stem Cell Research) (2i/L) and 1% Knockout Serum Replacement (KSR) on fibronectin at 37 °C in a 5% CO₂ humidified chamber. For serum/LIF (S/L) culture, cells were maintained on gelatin in GMEM supplemented with 10% FCS, ʟ-Glutamine, non-essential amino acids (NEAA), 0.1 mM β-mercaptoethanol, and 1000 U/ml LIF. Induction of EpiLC and PGCLC was carried out as previously described[12]. Briefly, ESCs cultured in 2i/L were washed thrice with PBS and seeded onto fibronectin (16.7 mg/ml) coated plates with EpiLC medium (N2B27, 1% KSR, bFGF (12 ng/ml), Activin A (20 ng/

ml)) for 42 h. Media was changed every day. EpiLC were then gently dissociated and seeded at 3000 cells per embryoid in ultra low-cell binding U-bottom 96-well plates (NUNC) with PGCLC induction medium (GK15: GMEM, 15% KSR, NEAA, 1 mM sodium pyruvate, 0.1 mM β-mercaptoethanol, 100 U/ml penicillin, 0.1 mg/ml streptomycin and 2 mM ʟ-glutamine; supplemented with cytokines: BMP4 (500 ng/ml), LIF (1000 U/ml), SCF (100 ng/ml), BMP8a (500 ng/ml) and EGF (50 ng/ml)). For large scale (CRISPR screen) PGCLC inductions (>4000 embryoids per replicate), 30 μl of PGCLC medium was used for each embryoid, with 20 μl additional PGCLC medium added on d2. For smaller-scale PGCLC inductions, 150 μl of PGCLC medium was used per embryoid and another 50 μl of PGCLC medium was added on d2. To minimise evaporation, only the inner 60 wells were used during the screen and 200 μl of PBS was added to the outer 36 wells. After 2–6 days, EBs were collected for FACS or gene expression analyses. SGET EpiLC were also seeded in GK15 medium without cytokines as a negative control to normalise for background levels of SGET fluorescence in embryoids. Where indicated, WNT inhibitor XAV939 (10 μM) or agonist CHIR99021 (3 μM) were added upon induction of PGCLC in embryoids. Where indicated DOX was added at 200 ng/μl upon induction of PGCLC in embryoids+/− above mentioned cytokines.

**Generation and validation of the SGET reporter**. Embryonic stem cell lines carrying *Stella*-GFP BAC (SG)[52] were re-derived as previously described, and routinely checked for mycoplasma-negative status using the mycoalert detection kit (Lonza). Low-passage SG lines were used to target an in-frame tdTomato cassette heterozygously into exon 1 of *Esg1* (ET) by homologous recombination, thereby generating SGET ESC. Targeting vector(s) for *Esg1* were constructed by amplifying homology arms from genomic DNA of mouse ESC by PCR using PrimeSTAR MAX (Takara Bio, Otsu, Japan), tdTomato from ptdTomato-N1 (Takara Bio, Otsu, Japan) and a destabilised domain (d2) with T2A BSD from pL1L2_IRESdCherryT2Abla-pAneotk vector (a gift from core facility in Cambridge Stem Cell Institute), and assembled using InFusion HD cloning kit (Takara Bio, Otsu, Japan). For gene targeting, 1 × 10⁷ low-passage ( < p10) *Stella*-GFP ESC suspended in PBS were electroporated with 40 μg linearised targeting vector using conditions: 800 V, 10 F 49 in Gene Pulser equipment (Bio-Rad). ESCs were selected with 10 mg/ml blasticidin, and colonies were picked and screened for correct targeting by PCR.

We tested for faithful reporter activity using tetraploid embryo complementation (~E8.5) and chimera formation assays (E12.5). For tetraploid complementation assay, 2-cell stage diploid embryos were collected in M2 medium from the oviduct at E1.5, and washed three times with medium containing 0.01% polyvinyl alcohol (Sigma), 280 mM Mannitol (Sigma), 0.5 mM Hepes (Sigma) and 0.15 mM MgSO4 (Sigma). Electrofusion of blastomeres to produce tetraploid embryos was subsequently carried out using a DC pulse (100 V/mm, 30 ms, 1 time) followed by application of AC pulses (5 V/mm, 10 s) using ECM 200 (BTX, Holliston, MA). Tetraploid embryos were transferred into KSOM medium (Merck Millipore) and cultured for 24 h for 4-cell/morula injection. For chimera formation assay, morula stage diploid embryos were collected in M2 medium from the oviduct of mice at E2.5. For micro-manipulation, a piezo-driven micro-manipulator (Prime Tech, Tokyo, Japan) was used to drill through the zona pellucida and 5–10 SGET ESC were introduced into the perivitelline space of morula stage tetraploid or diploid embryos. They were cultured to the blastocyst stage and then transferred into the uteri of pseudopregnant recipient MF1 female mice (E2.5). Post-implantation embryos were dissected at ~E4.5, E6.0, E7.0, E8.5 and E12.5 to analyse GFP and tdTomato expression. Notably these assays also confirm the authentic pluripotent status of our SGET line(s). All husbandry and experiments involving mice were authorised by a UK Home Office Project License 80/2637 and carried out in a Home Office-designated facility.

**Lentiviral CRISPR screen**. A genome-wide lentiviral CRISPR gRNA library was utilised that contains 87,897 gRNAs targeting 19,150 mouse protein-coding genes[21], with up to five gRNAs per gene (Addgene: #50947). The gRNA library was amplified with NEB 10-beta electrocompetent *Escherichia coli* (NEB) as per the recommended protocol. Briefly, *E. coli* were transformed via high-efficiency electroporation, to ensure faithful library replication, and incubated at 37 °C for 1 h with SOC recovery medium (ThermoFisher) before growing in 500 ml 2xTY (16 g/l Tryptone, 10 g/l Yeast Extract, 5.0 g/l NaCl) + ampicillin (50 μg/ml) medium at 37 °C overnight with 230 rpm shaking. Plasmid was purified from 500 ml bacterial cultures using an endotoxin-free plasmid maxi kit (Qiagen) as per manufacturer's instructions. Faithful replication/amplification of the library was confirmed by Illumina sequencing. For lentiviral production, 4 × 10⁶ HEK 293T cells cultured in DMEM (Invitrogen), 10% FCS, ʟ-glutamine and Penicillin-Streptomycin were seeded onto 10 cm poly-lysine coated plates. The following reagents were prepared in OptiMEM (ThermoFisher) per plate: 10 μg CRISPR plasmid library, 10 μg VSVG-pcDNA3 (Addgene:#8454), 10 μg pCMV-dR8.2 (Addgene:#8455) and 30 μl of Lipofectamine 2000 (ThermoFisher), and incubated with cells for 5–6 h at 37 °C in a 5% CO₂ incubator. Medium was subsequently replaced to include Forskolin (5 μM/ml), and viral supernatant was collected after 48 h and 64 h. Cell debris were removed by centrifugation at 1200 rpm for 15 min at 4 °C, and the supernatants were filtered with 0.45 μM filter. To concentrate the virus supernatant, it was combined with 4xPEG solution (320 ml of 50% PEG 6000, 40 ml of 5 M NaCl, 20

ml of 1 M HEPES, adjust pH to 7.4 and add ddH20 until 500 ml). The virus/PEG mixture was incubated for 4 h at 4 °C, centrifuged at $2600 \times g$ for 30 min at 4 °C, and the supernatant aspirated; the process was repeated twice. The viral titre (multiplicity of infection) was assayed by determining the percentage of cells with viral integration after transduction by flow analysing BFP-positive cells and confirmed by qPCR (for BFP).

For the CRISPR screen we first generated SGET ESC that carried constitutive expression of Cas9 by introducing a CAG-Cas9-pA cassette into SGET cells via PiggyBac-mediated transposition. Clones were subsequently picked and screened to identify a single-copy integration of Cas9, and to confirm functional capacity to direct high-efficiency gRNA-directed DNA indels using a GFP reporter. To introduce the CRISPR library, a total of $5 \times 10^7$ SGET-Cas9 ESC were transduced with lentiviral library using a multiplicity of infection (MOI) of 0.3, with biological replicates infected on independent occasions (gRNA fold coverage per replicate 146×; 296× total). Transduced SGET ESC were maintained on fibronectin-coated T175 flasks in N2B27, 2i/LIF and 1% KSR, with 8 ng/µl of polybrene added during lentiviral infection. Medium was changed 24 h later and cells were selected with puromycin (1.2 µg/ml) for 5 days. High numbers (>30mio) of transduced ESC were passaged every 2–3 days at low ratio to ensure the complexity of the gRNA library was maintained. ESC pellets were taken at day 12 for analysis and EpiLC were induced on day 12 post-lentiviral transduction, from which PGCLC were subsequently induced.

To quantify the relative frequencies of integrated gRNAs in each population during cell fate transitions, genomic DNA was isolated from day 12 ESC post-lentiviral transduction, from SG−ET+ and SG+ET+ EpiLC populations at 42 h, and from SG+ETlow PGCLCs from d6 embryoids, in biological duplicate using the DNeasy Blood & Tissue kit (Qiagen). Purified genomic DNA was used as template for custom primers that specifically amplify the gRNA region, and include overhanging Illumina adaptors and indexes to allow deep sequencing and multiplexing, respectively (synthesised as Ultramers from IDT) (see oligo tables). Prior to PCR, genomic DNA was sonicated with a Bioruptor (Diagenode) for 15 s on 'LOW' power to improve efficiency of amplification. gRNA sequences were amplified in multiple PCR reactions to enable all isolated DNA to be utilised, each with NEBNext® Q5 HotStart HiFi PCR Master Mix (NEB), 0.2 µM of universal forward primer (mix of staggers), 0.2 µM of indexed reverse primer and genomic DNA (~650 ng/50 µl rxn). For amplification from the plasmid vector, 15 ng was used as template for replicate amplifications. The following cycling conditions were used: 95 °C for 2 min, then between 21 and 26 cycles of (98 °C for 20 s, 62 °C for 20 s and 72 °C for 20 s) followed by 72 °C for 1 min and hold at 4 °C. The number of cycles was optimised for the point at which PCR products can be first visualised on an agarose gel. The PCR reaction was subsequently purified with AMPure® XP beads (Beckman Coulter) using double size selection to remove primer dimers and genomic DNA. Briefly, 0.55x of AMPure beads were added to PCR reaction (1×) and incubated for 5 min at room temperature. The supernatant was collected; the beads contained the unwanted larger fragments and were discarded. An additional 0.3× of AMPure beads was subsequently added to the supernatant (1×) and incubated for 5 min at room temperature. The supernatant was removed and beads washed twice with 200µl of 80% EtOH, air dried for 5 min and DNA was eluted from AMPure beads with EDTA-low TE buffer. The concentration of adaptor-ligated gRNA amplicons was measured with the Qubit DNA Assay kit (ThermoFisher Scientific) and the fragment distribution determined with an Agilent D1000 ScreenTape System (Agilent Technologies). Libraries were subsequently multiplexed and sequenced with an Illumina HiSeq 2500 using single-end 50 bp reads. As the sequences for all samples were identical up to the gRNA region, these types of low complexity libraries can produce low quality data. To counteract this, forward staggered primers were introduced at equal ratios during PCR amplification, generating offset reads. In addition, libraries were sequenced with four dark cycles and low density (70%) clustering.

## Gene editing in ESC cell lines

Targeted knockout ESC lines were generated using CRISPR genome editing by either deleting a critical exon to generate a frame-shifted null-allele or via inducing frame-shifting null indels into early coding exonic sequences. In brief, 20nt gRNAs encoding complementary sequences to the region/gene to be targeted were cloned into px459 (v2.0) (Addgene:#62988) using the Bbs1 sites and transfected into SGET ESC with lipofectamine 2000 (Thermo-Fisher), according to the manufacturers guidelines. A table of the gene targeting gRNAs and the editing strategy is shown below. Transfection of ESC was selected for with 1.2 µg/ml puromycin for 48 h, and ESC were subsequently passaged into 6-well plates at low density (1000–5000 ESC per well). One week later colonies were picked and expanded before undergoing genotyping by amplicon sizing or using the tracking of indels by decomposition tool (https://tide.nki.nl). Sanger sequencing on selected clones was subsequently used to confirm the precise mutant sequence of each allele. Multiple mutant lines were catalogued and validated before use in functional assays, where WT and/or heterozygous clones derived from the same targeting process were utilised as control, along with the parental SGET line. For doxycycline-inducible expression, Zfp296, Nr5a2 or Prdm14 cDNA was cloned into a custom PiggyBac vector downstream of a DOX-responsive promoter, and transfected into SGET ESC in conjunction with the reverse transactivator (TET-3G). Genomic integration was selected for with 250 µg/ml neomycin.

## Flow cytometry

For FACS, cultured cells or embryoids were dissociated into single cells with TrypLE, and suspended in phosphate buffered saline (PBS) with 0.5% BSA or 3% FBS. Filtered cell suspensions were subsequently flow sorted using a Sony SH800Z or a MoFlow high-speed cell sorter (Beckman Coulter) or analysed with BD LSRFortessa X-20 (BD Biosciences). For exit from pluripotency analyses, cells were sorted/analysed based on absence or presence of Stella-GFP expression, while maintaining Esg1-tdTomato activation (SG+/−ET+). For PGCLC, cells were sorted/analysed based on Stella-GFP re-activation coupled with low Esg1-tdTomato. The threshold(s) levels for sorting was set and normalised based on expression levels in pre-optimised SGET ESC. A negative population of ESC without any fluorescence was used to set absolute thresholds. Forward and side scatters were used to gate for the cell population and doublets. For each sort the maximum number of cells was collected, while for flow analysis at least 10,000 cell datapoints were captured to achieve a representative sample.

## RNA-seq and gene expression

For RNA-seq of $Nr5a2^{-/-}$, $Zfp296^{-/-}$ and matched WT control cell lines, EpiLC, d2 or d6 PGCLC were dissociated into single-cell solutions with TrypLE and sorted with a Sony SH800Z or a MoFlow high-speed cell sorter (Beckman Coulter) based on appropriate Stella-GFP and Esg1-tdTomato expression. Samples were collected in 150 µl of extraction buffer from the PicoPure RNA isolation kit (Life Technologies) and rapidly frozen on dry ice or liquid nitrogen. Total RNA was subsequently extracted using the PicoPure RNA isolation Kit, including a 15 min on-column DNaseI digestion. RNA integrity number was assessed with RNA HS ScreenTape (Agilent), and all samples confirmed to have a RIN > 8.5. 100 ng of total RNA from triplicate biological independent experiments was used as input for the NEBNext Ultra RNA library Prep Kit for Illumina® (NEB), with the library generated as per manufacturer's instructions. During the final PCR stage, 15 cycles of amplification were performed to generate the adaptor ligated, fragmented cDNA for sequencing. Samples were assessed using the High Sensitivity D1000 ScreenTape assay (Agilent) to ensure the library did not contain primer-dimer contamination after last round of AMPure beads cleaning. The Qubit dsDNA HS assay kit (ThermoFisher Scientific) and the NEBNext Library Quant Kit for Illumina® were used to accurately quantify the concentration of each library preparation. Samples were multiplexed with 12 indexes per lane and a total of two lanes of sequencing on an Illumina HiSeq2500 or HiSeq4000, single-end 50 bp, with an average depth of approximately 20 million reads per sample.

For RNA-seq from CRISPR screen samples, 50 ng of total RNA was used as input for the Ovation RNA-seq System v2 (Nugen) as per manufacturer's instructions. Amplified double-stranded (ds) cDNA was diluted into EDTA-low TE (Agilent) and sheared into ~230 bp in length using S220 Focused-Ultrasonicator (Covaris) using settings: duty factor 10%, cycle burst 200, intensity 5, temp at 4 °C and treatment time of 5 min per sample. A high Sensitivity D1000 ScreenTape assay (Agilent) was used to assess the efficiency of library fragmentation. Fragmented ds-cDNA was concentrated with Qiagen Reaction Clean Up kit (MiniElute) and 1 µg of the fragmented ds-cDNA used as input for the library preparation using Encore Rapid DR Multiplex Library System (Nugen). This kit ligated the adaptors to repaired-end ds-cDNA without amplification, to mimimise biases introduced during PCR amplification. The KAPA Library Quantification Kit (Kapa bioscience) was used to quantify the concentration of each adaptor-ligated libraries prior to multiplexing. The ESC, EpiLC and PGCLC RNA sequencing libraries were generated in parallel for each replicate, and the biological replicates were generated on independent occasions. Samples were multiplexed and sequenced with Illumina HiSeq1500, single-end 50 bp read length with a minimum depth of ~15 million reads per sample.

## DNA methylation

Global DNA methylation levels were determined using the LUminometric Methylation Analysis (LUMA) method[53] and bisulfite pyrosequencing. Briefly, genomic DNA was isolated from purified PGCLC using the DNeasy Blood & Tissue kit (Qiagen) and treated with RNase. 50–100 ng of DNA was digested with MspI/EcoRI and HpaII/EcoRI (NEB) and the subsequent methylation-sensitive overhangs were quantified by Pyrosequencing (PyroMark Q24 Advance) with the dispensation order: GTGTGTGTCACACAGTGTGT. Global CpG methylation levels were determined from relative peak heights at position(s) 7, 8, 13 and 14 using the formula: $[(2*(p7*p13))/(p8 + p14)]^{HpaII}/[(2*(p7*p13))/(p8 + p14)]^{MspI}$. CpG methylation at LINE1 loci was determined by bisulfite pyrosequencing using the EpiTect bisulfite kit (Qiagen). PCR amplification and assay design were performed as previously described[54].

## Bioinformatics

For RNA-seq, expression reads were quality-trimmed using TrimGalore to remove adaptors, and aligned to the mouse reference genome (GRCm38/mm10) using TopHat2 guided by ENSEMBL gene models. Mapped reads were imported into SeqMonk analysis software, normalised, and quantitated using the RNA-seq quantitation pipeline. Differentially expressed genes (DEG) were identified using the DESeq2 statistical package with fdr significance thresholds set as $p < 0.05$, filtering for a fold-change (FC) > 2 and mean RPKM expression > 1 in at least one sample. Analysis of gene ontology enrichment was performed using DEG datasets in the DAVID bioinformatics resource v6.7 (https://david.ncifcrf.gov) and/or the REVIGO visualisation tool (http://revigo.irb.hr). Heatmaps and

plots were generated using custom scripts in the R statistical package. For analysis of the CRISPR screen, demultiplexed gRNA sequences were extracted from Illumina reads using custom scripts to account for staggered start positions. These were quantitated and statistically analysed using model-based analysis of genome-wide CRISPR–Cas9 knockout (MAGeCK) v.0.5.2 software in python[22]. Library sequences were trimmed to exclude the bottom 1% of reads from the initial library, which primarily corresponded to zero-count gRNAs. Fold-changes in gRNA frequency were calculated as the logarithmic of the mean fold-change between each gRNA set(s) that target a common gene between paired fate transitions. Statistical analysis to identify significantly affected genes was performed using a 'total' normalisation method in MAGeCK to compare gRNA counts with the preceding cell population during ESC-EpiLC-PGCLC transit (or with the vector library in the case of ESC). We subsequently applied the relative ranking algorithm, taking genes scoring $p < 0.01$ for negative or positive enrichment in both independent replicate screens as candidates for further validation.

## Data availability

All raw sequencing data related to this study has been deposited in the gene expression omnibus (GEO) repository under accession GSE117473. Other relevant data and reagents that support the findings of this study are available from the corresponding author(s) on reasonable request. Information on oligonucleotide sequences can be found in the supplementary material.

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

## Acknowledgements
We thank Andy Riddell, Charles Bradshaw and all members of the M.A.S. and J.A.H labs for discussions and support. Funding for this study came from a Wellcome Trust programme grant (C6946/A14492) and Cancer Research UK/Wellcome Trust (092096) core grants to M.A.S, and from a core European Molecular Biology Laboratory (EMBL) grant to J.A.H. The Newton Trust and a Leverhulme Trust Early Career fellowship supported U.G. The James Baird Fund supported Y.H.

## Author contributions
J.A.H. designed the study, performed experiments and bioinformatics, and wrote the manuscript. Y.H. performed experiments and bioinformatics. U.G. performed experiments and wrote the manuscript. K.H.-G. performed experiments. T.K. performed experiments. M.A.S. supervised the study and wrote the manuscript.

## Additional information

**Competing interests:** The authors declare no competing interests.

