## [Peer Review file · Nature Communications]

Reviewers' comments:

Reviewer #1 (Remarks to the Author):

Hackett et al. perform a genome-wide CRISPR screen to identify essential factors for differentiation of primordial germ stems from pluripotent mouse embryonic stem cells. They perform a 2-step differentiation with a dual-color reporter for each stage and look for gene knockouts that block successful differentiation. They identify several factors and validate two of these — Zfp296 and Nr5a2 — using single guides to create isogenic stem cell lines.

The study is well performed but I have a few minor comments that should be addressed:

Screen representation: Details should be added on the representation of the screen (cells per guide RNA) and whether infection replicates were used.

FACS details: Was a full screen representation sorted? Was a full screen representation collected for each population? It is important to provide these details to understand if screen representation was adequately maintained.

Statistical methods for pooled screen gRNA significance need to be described in more detail. It would help if the authors could provide more transparent metrics (rather than just p value, which is given without any test specific). For example, were all the gRNAs enriched above some non-targeting controls? For that matter, did the screen include non-targeting controls?

To identify the targets of Nr5a2 and Zfp296, it might be helpful to perform a ChIP assay and see if these targets are also enriched in the genome-wide screen.

Reviewer #2 (Remarks to the Author):

The authors developed a mouse embryonic stem cell (mESC) compound-reporter system for developmental transitions towards PGC fate in vitro (Stella-GFP and Esg1-tdTomato). They used tetraploid complementation and chimera formation to prove that the system faithfully recapitulates the expression of endogenous Stella and Esg1 during the transitions towards PGC fate. Next the authors functionally validated the reporter system by generating Blimp1 mutant in this system and showed that the PGC-like cell (PGCLC) generation efficiency is strongly decreased. Then they used CRISPR system to screen a guide RNA library and identified 627 candidate genes that lead to loss of naïve state when being knocked out. These candidates include Oct4 and Sox2, validating their effectiveness. They also identified 21 candidate genes (mainly tumor suppressor genes) that lead to resistance to exit from naïve pluripotency when being knocked out. The authors chose naïve state specific candidate Ogdh and used siRNAs to knock it down to prove the phenotype. They also chose naïve-prime state shared candidate Txn1 to use siRNAs to knock it down to prove the phenotype.

Then the authors screened for the genes promoting the dissolution of naïve pluripotency and identified 21 enriched candidate genes including Tcf3, Zfp281, Dnmt1, and Rest. They individually knocked out Zmym2, FoxP1, Uchl5, and Zfp281 and confirmed their phenotypes.

Thirdly the authors screened for the genes important for specification of PGC fate. They identified 23 candidate genes involved in PGC specification and/or nascent PGC development. Then they focused on Zfp296 and Nr5a2 and individually knocked them out. Nr5a2 knockout mESCs showed marked

reduction of PGCLC specification efficiency whereas Zfp296 knockout mESCs showed marked reduction of day6 PGCLCs. The authors further showed that Nr5a2 knockout lead to aberrant transduction of WNT signaling in PGCLC and upregulation of mesendoderm master genes and de-repression of somatic cell program. On the contrary, Zfp296 knockout leads to severe block of WNT pathway genes. Then the authors showed that adding WNT inhibitor XAV939 led to strong downregulation of key PGC genes, consistent with the effects of impaired WNT signaling in Zfp296 knockout PGCLC. Adding WNT agonist Chiron led to upregulation of T and Wnt3 and downregulation of Nanog, consistent with the effects of over-activity of WNT signaling in Nr5a2 knockout PGCLC. Finally the authors showed that addition of WNT inhibitor to Nr5a2 knockout PGCLC rescued the hyper-activation of T and Wnt3. But addition of WNT agonist to Zfp296 knockout PGCLC only resulted in modest rescue of aberrantly repressed WNT targets.

The work is elegantly designed, properly performed and independently validated. The writing is accurate and elegant. The result is very important for the field and it should be accepted for publication in NC.

Reviewer #3 (Remarks to the Author):

In this manuscript, Hackett et al. combined a compound-reporter Stella-GFP and Esg-tdTomato (SGET) system, with lentivirus based genome-wide CRISPR/Cas9 screen to search for critical regulators in mouse PGCLC derivation. They identified not only important genes for the maintenance of, and exit from the mouse naive pluripotent state, but also critical genes for epiblast fate acquisition, as well as PGC fate specification. They identified two genes, Nr5a2 and Zfp296, as being important for mouse PGC fate determination, with further functional studies showing that deletion of Nr5a2 induced overactivation of Wnt pathway, while loss of Zfp296 caused failure of Wnt pathway activation thus resulting in the incompetence of PGCLC induction.

Specific Comments

1. A previous study used BVSC (Blimp1-Venus and Stella-CFP) mouse ESCs to differentiate PGCLC in vitro, they reported that Stella-CFP is repressed in mouse ESCs, which only turned on in PGCLC stage; in contrast, in this manuscript, the mouse ESC is highly positive for Stella-GFP, can the authors explain why the transgene expression is different from earlier studies?
2. In Figure 1C, the authors define the SG+ETlow population as PGCLC, however we can still see some SG+ET+ cells in the flow chart, can the authors explain what those cells are? Is it possible that they are also PGCLCs?
3. The logic of the CRISPR/Cas9 screen is to compare the gRNA frequency to the preceding cell population, however, this strategy excludes genes that are important to the survival of both PGCLC and the preceding ESCs/EpiLCs, because the deleted cells will die in the early stages. Have the authors considered using an inducible CRISPR/Cas9 to screen for genes that are more specific for PGC fate determination?
4. In the manuscript, the authors mentioned the GO analyses of differentially expressed genes of both Nr5a2-null and Zfp296-null day2 PGCLC respectively, but I could not find the data?
5. In the heatmap Figure 5A, Stra8 is unexpectedly highly upregulated in d2 ZFP296-/- PGCLC compared to d2 WT, can the authors give some explanation for this result? In the manuscript the authors also mentioned the delayed upregulation of PGC-related genes in ZFP296-/- cells, have the

authors compared the PGCLC derivation efficiency in later stages, like d8/d10/d12 in ZFP296-/- PGCLC?

6. The authors suggested that Nr5a2 plays a direct role in regulating Wnt pathway in PGCLC development, while the detailed mechanism is unclear. Any additional insights?

Response to reviewer comments

Reviewer #1

Hackett et al. perform a genome-wide CRISPR screen to identify essential factors for differentiation of primordial germ stems from pluripotent mouse embryonic stem cells. They perform a 2-step differentiation with a dual-color reporter for each stage and look for gene knockouts that block successful differentiation. They identify several factors and validate two of these — Zfp296 and Nr5a2 — using single guides to create isogenic stem cells lines.

We thank the reviewer for their time spent on our manuscript and the constructive comments.

The study is well performed but I have a few minor comments that should be addressed:

1. Screen representation: Details should be added on the representation of the screen (cells per guide RNA) and whether infection replicates were used.

Our study was performed using independently infected replicates, followed by independent phenotypic read-out experiments (PGCLC specification). Moreover, within each biological replicate infection there are typically 4-5 independent gRNAs targeting each gene, which act as additional internal replicates. We infected 50 million ESC in each biological replicate at MOI of 0.3, giving a library fold-coverage of 147x (12.9mio cell/87,897 gRNA), or 296x across replicates. These details are included in the methods section but have now been more explicitly stated.

2. FACS details: Was a full screen representation sorted? Was a full screen representation collected for each population? It is important to provide these details to understand if screen representation was adequately maintained.

To maintain adequate library coverage, we seeded a minimum of 30mio cells during experimental passaging and differentiation. Analysis of each independent replicate of ESC and EpiLC was performed using DNA corresponding to approximately 10mio unique cells, whilst fewer cells were used for PGCLC owing to inherent limitations in efficiently generating this cell-type, as shown in supplementary figure 2 and discussed in the manuscript (Line 147-149). We have modified the methods to further discuss library representation in our study.

3. Statistical methods for pooled screen gRNA significance need to be described in more detail. It would help if the authors could provide more transparent metrics (rather than just p value, which is given without any test specific). For example, were all the gRNAs enriched above some non-targeting controls? For that matter, did the screen include non-targeting controls?

We used the Model-based Analysis of Genome-wide CRISPR-Cas9 Knockout (MAGeCK) tool to ascertain statistically significant *p-values* at the level of gene KO enrichment or depletion¹. This highly cited method first determines a *p-value* for the enrichment of each gRNA relative to control based on dispersion from the mean in a negative binomial distribution model. It subsequently uses the robust ranking algorithm (RRA) to determine whether multiple gRNAs targeting the same gene have concordantly shifted within the distribution of *p-values*, thereby implying significant enrichment/depletion. We have now highlighted more clearly throughout the manuscript which statistical tests were used to calculate *p-values*.

The gRNA library used was that reported by Yusa *et al.*². This library does not include non-targeting gRNAs, but importantly, we can utilise the 1,000s of gRNAs that target genes which are either not expressed or are known not to have a critical function in PGCs as equivalent negative controls. For example, in supplementary figure 4C, we show that gRNAs targeting any gene in *Hoxa* and *Slc25a* gene family clusters have no effect in our screen. Importantly, of the key candidates we identified, we carry out a thorough validation and dissect their mechanism of action, thereby underscoring the true-positive efficacy of the screen/library.

4. To identify the targets of *Nr5a2* and *Zfp296*, it might be helpful to perform a ChIP assay and see if these targets are also enriched in the genome-wide screen.

We reasoned that to identify the functional targets of *Nr5a2* and *Zfp296*, loss- and gain- of function studies would be advantageous since they indicate the direct transcriptional outputs regulated by these proteins rather than solely binding sites, which can have unknown targets (e.g. when in *cis*-regulatory regions). Indeed, our extensive genetic and transcriptomic analysis revealed the core set of genes responsive to *Nr5a2* or *Zfp296* in germ cells, thereby delineating key functional pathways regulated by these genes. By contrast, ChIP-seq will most likely reveal 100s-1000s binding sites and it will be difficult to identify critical (functional) binding sites. Moreover, it is highly challenging to generate sufficient *in vitro* PGCs to perform ChIP for transcription factors, especially given the relatively poor antibodies available. Finally, binding targets will not necessarily be enriched in the functional screen, especially if they act extrinsically (e.g. WNT proteins). Taken together we therefore believe such an approach would not significantly enhance our conclusions, which are based on robust genetics coupled with transcriptomics.

Reviewer #2 :

The authors developed a mouse embryonic stem cell (mESC) compound-reporter system for developmental transitions towards PGC fate in vitro (Stella-GFP and Esg1-tdTomato). They used tetraploid complementation and chimera formation to prove that the system faithfully recapitulates the expression of endogenous Stella and Esg1 during the transitions towards PGC fate. Next the authors functionally validated the reporter system by generating Blimp1 mutant in this system and showed that the PGC-like cell (PGCLC) generation efficiency is strongly decreased. Then they used CRISPR system to screen a guide RNA library and identified 627 candidate genes that lead to loss of naïve state when being knocked out. These candidates include Oct4 and Sox2, validating their effectiveness. They also identified 21 candidate genes (mainly tumor suppressor genes) that lead to resistance to exit from naïve pluripotency when being knocked out. The authors chose naïve state specific candidate Ogdh and used siRNAs to knock it down to prove the phenotype. They also chose naïve-prime state shared candidate Txn1 to use siRNAs to knock it down to prove the phenotype.

Then the authors screened for the genes promoting the dissolution of naïve pluripotency and identified 21 enriched candidate genes including Tcf3, Zfp281, Dnmt1, and Rest. They individually knocked out Zmym2, Foxp1, Uchl5, and Zfp281 and confirmed their phenotypes.

Thirdly the authors screened for the genes important for specification of PGC fate. They identified 23 candidate genes involved in PGC specification and/or nascent PGC development. Then they focused on Zfp296 and Nr5a2 and individually knocked them out. Nr5a2 knockout mESCs showed marked reduction of PGCLC specification efficiency whereas Zfp296 knockout mESCs showed marked reduction of day6 PGCLCs. The authors further showed that Nr5a2 knockout lead to aberrant transduction of WNT signaling in PGCLC and upregulation of mesendoderm master genes and de-repression of somatic cell program. On the contrary, Zfp296 knockout leads to severe block of WNT pathway genes. Then the authors showed that adding WNT inhibitor XAV939 led to strong downregulation of key PGC genes, consistent with the effects of impaired WNT signaling in Zfp296 knockout PGCLC. Adding WNT agonist Chiron led to upregulation of T and Wnt3 and downregulation of Nanog, consistent with the effects of over-activity of WNT signaling in Nr5a2 knockout PGCLC. Finally the authors showed that addition of WNT inhibitor to Nr5a2 knockout PGCLC rescued the hyper-activation of T and Wnt3. But addition of WNT agonist to Zfp296 knockout PGCLC only resulted in modest rescue of aberrantly repressed WNT targets.

The work is elegantly designed, properly performed and independently validated. The writing is accurate and elegant. The result is very important for the field and it should be accepted for publication in NC.

We thank the reviewer for their time and effort in reviewing our manuscript.

Reviewer #3 :

In this manuscript, Hackett et al. combined a compound-reporter Stella-GFP and Esg-tdTomato (SGET) system, with lentivirus based genome-wide CRISPR/Cas9 screen to search for critical regulators in mouse PGCLC derivation. They identified not only important genes for the maintenance of, and exit from the mouse naive pluripotent state, but also critical genes for epiblast fate acquisition, as well as PGC fate specification. They identified two genes, Nr5a2 and Zfp296, as being important for mouse PGC fate determination, with further functional studies showing that deletion of Nr5a2 induced overactivation of Wnt pathway, while loss of Zfp296 caused failure of Wnt pathway activation thus resulting in the incompetence of PGCLC induction.

We thank the reviewer for their time and constructive comments.

Specific Comments

1. A previous study used BVSC (Blimp1-Venus and Stella-CFP) mouse ESCs to differentiate PGCLC in vitro, they reported that Stella-CFP is repressed in mouse ESCs, which only turned on in PGCLC stage; in contrast, in this manuscript, the mouse ESC is highly positive for Stella-GFP, can the authors explain why the transgene expression is different from earlier studies?

The previously reported *Stella*-CFP (BVSC) line³ used a short upstream regulatory fragment of *Stella* to drive expression of the reporter gene (CFP). As noted by the reviewer this specific reporter is not active in mouse ESC, only in PGCLC. In contrast our *Stella*-GFP reporter utilises the full set of *Stella* regulatory elements and is active both in ESC and PGCLC. Crucially, endogenous *Stella* is highly active in both ESC (in 2i/L) and PGCLC, as well as their in vivo equivalents, and therefore our reporter faithfully recapitulates the appropriate *Stella* expression pattern (see Figure 1A). Indeed, we fully validate our reporter system *in vivo* to confirm it faithfully reflects the expression pattern of the endogenous genes in an embryonic context (see Figure 1A & B). Thus, our SGET reporter more accurately recapitulates endogenous gene expression patterns, and thereby represents the authentic changes in cell state transitions.

2. In Figure 1C, the authors define the SG+ETlow population as PGCLC, however we can still see some SG+ET+ cells in the flow chart, can the authors explain what those cells are? Is it possible that they are also PGCLCs?

We have analysed the small population of SG+ET+ cells that emerge during PGCLC derivation, and found that indeed they express some PGCLC markers (e.g. *Prdm14*, *Nanos3*) as hypothesised by the reviewer, albeit at lower levels. However this population also expressed appreciably higher levels of some ESC-specific genes (e.g. *Klf4*), consistent with SG+ET+ ESC-like identity. This implies that this population includes both PGCLC and a small number of reverting ESC-like cells. We therefore chose to exclude them from analysis to avoid confounding effects that would reduce statistical power to identify true PGC-specific regulators in the screen. Indeed, this observation underscores that the SGET system allows us to filter for authentic PGCLC fate, which is particularly important for any *in vitro* differentiation system which are inherently susceptible to heterogeneity.

3. The logic of the CRISPR/Cas9 screen is to compare the gRNA frequency to the preceding cell population, however, this strategy excludes genes that are important to the survival of both PGCLC and the preceding ESCs/EpiLCs, because the deleted cells will die in the early stages. Have the authors considered using an inducible CRISPR/Cas9 to screen for genes that are more specific for PGC fate determination?

The reviewer is correct in that any essentiality genes for e.g. ESC cannot subsequently be assayed for its role in PGCLC. We note however that 648 of 19,149 genes were scored as essential in our screen in ESC/EpiLC - with most being involved core processes common to all cell-types - and thereby we still achieve near-genome wide coverage for PGC regulators. Notably, because we use 2i/L ESC culture media, which can buffer against perturbation of the pluripotency network, we are still able to analyse knock-outs for most pluripotency-associated genes in the germline, since KO ESC are viable.

The suggestion to perform the screen with inducible Cas9 is excellent, but implementing this approach is limited by technical requirements. Specifically, evidence has shown it takes a minimum of 7 days, and more appropriately >10 days, to ensure all genes in the population carry targeted knockouts after induction/transduction of Cas9 and gRNA expression⁴. Ensuring that all/most cells that carry a gRNA are also knockout is necessary to avoid confounding effects, particularly in a dropout screen. Because EpiLC and PGCLC induction occurs over the first 2 or 6 days respectively, an inducible system would fail to generate a sufficient proportion of knock-out cells prior to acquisition of cell identity, thereby blunting

statistical power to detect depletion/enrichment effects. For this reason, inducible Cas9 screens cannot be performed over such short periods of cell fate transitions.

4. In the manuscript, the authors mentioned the GO analyses of differentially expressed genes of both Nr5a2-null and Zfp296-null day2 PGCLC respectively, but I could not find the data?

We have now included a supplementary file including the GO analysis of the transcriptomics data

5. In the heatmap Figure 5A, Stra8 is unexpectedly highly upregulated in d2 ZFP296-/- PGCLC compared to d2 WT, can the authors give some explanation for this result? In the manuscript the authors also mentioned the delayed upregulation of PGC-related genes in ZFP296-/- cells, have the authors compared the PGCLC derivation efficiency in later stages, like d8/d10/d12 in ZFP296-/- PGCLC?

One possibility is that putative d2 *Zfp296* KO PGCLC have acquired an alternative cell fate, as judged by failure to upregulate early germ cell genes, and that *Stra8* expression is indicative of a general mis-regulated transcriptional programme. Alternatively, *Stra8* upregulation could be directly linked with impaired WNT activity in *Zfp296* KO PGCLC. Indeed reciprocally, *Stra8* is downregulated in *Nr5a2* KO PGCLC (d6) that had hyperactive WNT, thereby implying *Stra8* could be a WNT target in the germline. Regarding PGCLC derivation efficiency at later timepoints, we have not examined specification beyond d6, albeit it is unlikely that after d3-4 there is further specification of PGCLC, since the state of competence (EpiLC) is transient³. Thus, any effect observed by d6 is likely to similarly propagate through to later timepoints.

6. The authors suggested that Nr5a2 plays a direct role in regulating Wnt pathway in PGCLC development, while the detailed mechanism is unclear. Any additional insights?

This is an excellent point. It is possible *Nr5a2* impinges Wnt signalling in the germline through either direct or indirect mechanisms that likely will require complex controlled experiments and genetics to disentangle. Our assessment is that *Nr5a2* may influence PGCLC ability to respond to WNT signalling via negative feedback loops rather than directly effecting WNT transduction, albeit it will be important in future to dissect the exact mechanism(s) involved. Indeed, our data indicates that precisely regulated levels of WNT are necessary to ensure robust PGCLC specification. However, understanding such a broad and complex process is beyond the scope of this study.

1. Li, W. *et al.* MAGeCK enables robust identification of essential genes from genome-scale CRISPR/Cas9 knockout screens. *Genome Biol* **15**, 554 (2014).
2. Koike-Yusa, H., Li, Y., Tan, E.P., Velasco-Herrera Mdel, C. & Yusa, K. Genome-wide recessive genetic screening in mammalian cells with a lentiviral CRISPR-guide RNA library. *Nat Biotechnol* **32**, 267-73 (2014).
3. Hayashi, K., Ohta, H., Kurimoto, K., Aramaki, S. & Saitou, M. Reconstitution of the mouse germ cell specification pathway in culture by pluripotent stem cells. *Cell* **146**, 519-32 (2011).
4. Shalem, O. *et al.* Genome-Scale CRISPR-Cas9 Knockout Screening in Human Cells. *Science* **343**, 84-87 (2014).

REVIEWERS' COMMENTS:

Reviewer #1 (Remarks to the Author):

My concerns have been addressed in the revised manuscript.

Reviewer #3 (Remarks to the Author):

Our comments and concerns have been addressed by the authors

Response to reviewer comments

Reviewer #1 (Remarks to the Author):

My concerns have been addressed in the revised manuscript

We thank the reviewer for their time spent on our manuscript and the constructive comments during the course of review. We note that the reviewer is satisfied with our revised manuscript.

Reviewer #3 (Remarks to the Author):

Our comments and concerns have been addressed by the authors

We thank the reviewer for their time spent on our manuscript and the constructive comments during the course of review. We note that the reviewer is satisfied with our revised manuscript.